# Statins and Colorectal Cancer Risk: A Population-Based Case-Control Study and Synthesis of the Epidemiological Evidence

**DOI:** 10.3390/jcm11061528

**Published:** 2022-03-10

**Authors:** Antonio Rodríguez-Miguel, Encarnación Fernández-Antón, Diana Barreira-Hernández, Luis A. García-Rodríguez, Miguel Gil, Alberto García-Lledó, Francisco J. De Abajo

**Affiliations:** 1Clinical Pharmacology Unit, University Hospital “Príncipe de Asturias”, 28805 Alcalá de Henares, Spain; antonio.hupa@gmail.com (A.R.-M.); encarny29@gmail.com (E.F.-A.); dbarreirahdez@gmail.com (D.B.-H.); 2Pharmacology Unit, Department of Biomedical Sciences, School of Medicine and Health Sciences, University of Alcalá (IRYCIS), 28805 Alcalá de Henares, Spain; 3Spanish Centre for Pharmacoepidemiologic Research (CEIFE), 28004 Madrid, Spain; lagarcia@ceife.es; 4BIFAP Unit, Pharmacoepidemiology and Pharmacovigilance Division, Spanish Agency for Medicines and Clinical Devices (AEMPS), 28022 Madrid, Spain; mgilg@aemps.es; 5Department of Cardiology, University Hospital “Príncipe de Asturias”, 28805 Alcalá de Henares, Spain; josealberto.garcia@salud.madrid.org; 6Department of Medicine and Medical Specialties, School of Medicine and Health Sciences, University of Alcalá, 28805 Alcalá de Henares, Spain

**Keywords:** statin, colorectal cancer, chemoprevention, real-world data, real-world evidence

## Abstract

(1) Background: The pleiotropic effects of statins may explain a chemoprotective action against colorectal cancer (CRC). Many studies have tested this hypothesis, but results have been inconsistent so far. Moreover, few have examined statins individually which is important for determining whether there is a class effect and if lipophilicity and intensity may play a role. (2) Methods: From 2001–2014, we carried out a study comprised of 15,491 incident CRC cases and 60,000 matched controls extracted from the primary healthcare database BIFAP. We fit a logistic regression model to compute the adjusted-odds ratios (AOR) with their 95% confidence intervals (CIs). Additionally, we carried out a systematic review and meta-analysis. (3) Results: Current use of statins showed a reduced risk of CRC (AOR = 0.87; 95% CI: 0.83–0.91) not sustained after discontinuation. The association was time-dependent, starting early (AOR_6months–1year_ = 0.85; 95% CI: 0.76–0.96) but weakened beyond 3-years. A class effect was suggested, although only significant for simvastatin and rosuvastatin. The risk reduction was more marked among individuals aged 70 or younger, and among moderate-high intensity users. Forty-eight studies were included in the meta-analysis (pooled-effect-size = 0.90; 95% CI: 0.86–0.93). (4) Conclusions: Results from the case-control study and the pooled evidence support a moderate chemoprotective effect of statins on CRC risk, modified by duration, intensity, and age.

## 1. Introduction

In 2020, colorectal cancer (CRC) was the third type in number of new cases worldwide and the first most incident in Europe, after excluding gender-specific types [1]. Better treatments, together with the spread of population-wide screening policies, have improved survival, although their effectiveness critically depends on the stage at diagnosis [2,3]. Almost 80% of diagnosed CRCs have no identifiable genetic basis [2,3] and arise from pre-existing polyps, in which pro-inflammatory factors [4,5], including hypercholesterolemia [6,7], seem to play a role [2,3,4,5,6,7]. Chemoprevention of CRC acting on these factors has emerged as a promising strategy [8]; in fact, this was evidenced in 2016 when the U.S. Preventive Services Task Force (USPSTF) endorsed the use of low-dose aspirin for the primary prevention of cardiovascular disease (CVD) and CRC in a specific population [9]. However, they recently revised and downgraded that statement [10] once the net benefit of low-dose aspirin for primary prevention of CVD was put into question [11,12,13]. Along these lines, statins, of which among their pleiotropic effects include anti-inflammatory and anti-proliferative actions [14], have been widely studied in observational and post-hoc analyses from randomized clinical trials as potential chemoprotective agents against CRC, but no conclusive results have been reached; this is mostly due to the high heterogeneity across studies [15]. Furthermore, there are still a number of questions that remain unclear; for instance, in most of the studies published to date, authors assumed homogeneous effects across all statins against CRC, however, there is scarce data available regarding individual drugs to support that assumption (low numbers of many of them being a limitation for that purpose) [15]. This is not a trivial issue because extrahepatic effects of statins may be linked to their lipophilicity [16] and intensity of use (with regards to active principle and daily dose) [16], so both features can only be addressed by studying the association with CRC at individual drug level. Finally, the interaction with other concomitant drugs used for CVD prevention, such as low-dose aspirin, is a gap previously pointed out by the USPSTF [9] that also remains unexplored.

Thus, the main objective of the present study was to examine the chemoprotective effect of statins on CRC as a whole group and by active principle, which in turn would allow us to test whether such is a class effect and whether it is dependent on their lipophilicity and intensity. As a secondary objective we examined the potential synergistic effect of statins with other drugs used for CVD prevention, as antiplatelet drugs. Finally, we synthesized the available epidemiological evidence in a systematic review with meta-analysis.

## 2. Subjects and Methods

### 2.1. Source of Data

We carried out a case-control study nested in a cohort extracted from BIFAP (“Base de datos para la Investigación Farmacoepidemiológica en Atención Primaria”) [17], a healthcare database containing pseudonymized electronic health records of patients attended by primary care physicians (PCPs) from 10 Spanish regions (out of 17). The population included in BIFAP is representative of the Spanish population, and the database has been successfully validated by comparison to other well-known European databases for pharmacoepidemiologic research purposes [17]. BIFAP comprises information on demographics, medical problems, and drug prescriptions (product name, dosage, indication, date, and duration of supply), among many other data [17]. We used BIFAP in its 2014 version, which included 7.6 million patients and a 5.1 year follow-up on average per subject (a total of 38.6 million person-years).

### 2.2. Design

Through the period from 1 January 2001 to 31 December 2014, subjects fulfilling the inclusion criteria (aged 20–89, without record of any type of cancer, and with at least 1-year record with their PCP) conformed the study cohort and the date of entering was the “start date”. All of the subjects from the study cohort (*n* = 5,310,198) were then followed up until the occurrence of the event of interest (an incident CRC), a censoring event (a cancer diagnosis different from CRC, 90 years old or death), or the end of the study period, whichever came first.

### 2.3. Selection of Cases and Controls

Previously, we successfully validated the CRC diagnosis in BIFAP in a study published elsewhere [18]. Briefly, we built an algorithm to mine codes from the International Classification of Primary Care (ICPC-2) and the International Classification of Diseases (ICD-9) with their linked text descriptors to search for potential cases of CRC [18]. Potential cases retrieved were considered valid when we found additional supporting information in their healthcare records, such as specialist referrals, tumor information (diagnostic procedures, location, histopathology, staging or treatments) or confirmed death from CRC [18]. We excluded all of the subjects with a diagnosis of CRC of known genetic origin.

We selected controls using an incidence-density sampling [19]. To this end, we assigned a random date within the study period to all of the subjects in the cohort, so they were considered as eligible controls when the random date fell within their period of observation. This way we ensured that the more person-time a subject contributes to the cohort, the higher the probability to be selected as a potential control [19]. Finally, from the pool of candidates, we sampled without replacement 60,000 controls frequency-matched to cases by age, sex, and year of event occurrence. The “index date” for cases was the first recorded CRC diagnosis and for controls was the random date assigned before sampling.

### 2.4. Drugs Included and Exposure Definition

Statins were studied as a group and individually, including simvastatin, atorvastatin, pravastatin, lovastatin, fluvastatin, rosuvastatin, and pitavastatin. Statins were grouped into hydrophilic (rosuvastatin and pravastatin) and lipophilic (the rest), and also into three categories of intensity (low, moderate and high) depending on the targeted reduction in blood LDL-cholesterol achievable by dose and active principle [20].

Recency of use of statins in cases and controls was ascertained from the index date backwards and classified as: *current users* when the supply of the last recorded prescription ended between 0–90 days; *recent users* when finished between 91–365 days; *past users* when finished beyond 365-days; *non-users* if no prescriptions were recorded in the database.

Treatment duration of statins was assessed after adding up the duration of all continuous prescriptions, considered as that when the gap between the end of supply of one and the beginning of the following were equal or lower than 90-days. We used the same definitions of exposure for all of the drugs studied.

### 2.5. Confounding Assessment

Potential confounders (including co-medications) and risk factors were selected by domain knowledge and ascertained any time before the index date. We included the following comorbidities and risk factors: chronic gastritis, gastro-esophageal reflux, inflammatory bowel disease, irritable bowel syndrome, complicated upper gastrointestinal (GI) disorders (complicated ulcer, bleeding gastritis or duodenitis and upper GI bleeding), non-complicated upper GI disorders (non-bleeding or non-complicated ulcer, gastritis or duodenitis), dyspepsia, lower GI bleeding, constipation, anorectal pathology (hemorrhoids, anal fissure and anorectal abscess), alcohol abuse, smoking status, body mass index (BMI), hyperuricemia, gout, and the number of visits to the PCP in year before the index date. We also included the use of the following co-medications: nonsteroidal anti-inflammatory drugs (NSAIDs), symptomatic slow acting drugs for osteoarthritis (SYSADOAs), corticosteroids, low-dose aspirin and non-aspirin antiplatelet drugs, oral anticoagulants, oral glucose-lowering drugs and insulin, serotonin selective reuptake inhibitors, serotonin receptor antagonists with serotonin reuptake inhibition, drugs for peptic ulcer and gastro-esophageal reflux (proton-pump inhibitors and H2-receptor antagonists), anti-diarrheal drugs, analgesic opioids, drugs for constipation, antihypertensives, other lipid-lowering drugs (fibrates, bile acid sequestrants and other lipid-modifying agents), and calcium with or without vitamin D supplements.

### 2.6. Statistical Analysis

To estimate the association between the exposure of interest and the risk of CRC, we specified an unconditional logistic regression model to compute the unadjusted odds-ratios (OR; including only the matching variables as predictors), and the fully adjusted-odds ratios (AORs; including the matching variables and the vector of above-mentioned confounders as predictors), with their 95% confidence intervals (95% CI). The incidence-density sampling method of controls ensures that the OR obtained is an unbiased estimate of the incidence rate ratio in the underlying cohort, even in presence of competing risks [19,21].

With a sample size of 15,491 CRC cases and 60,000 matched controls, a targeted AOR of 0.80 or lower, a confidence level of 95% in a bilateral test, we would reach a statistical power of at least 85%, provided the prevalence of use of statins among controls was 1.5% or higher.

The variables such as “smoking” and “BMI” were not systematically recorded by PCPs in their daily routine, so there were missing data at the time of ascertainment. After confirming the pattern of missingness was “missing at random”, we fit a multiple imputation by chained equations (MICE) model to obtain 20 imputed databases then used in the logistic model to compute all measures of association [22].

We evaluated the interaction between statins and age (≤70, >70 years), sex, and BMI in a multiplicative scale and tested using the method proposed by Altman and Bland [23]. We also evaluated the interaction between statins and antiplatelet drugs (including low-dose aspirin), antihypertensives, other lipid-lowering drugs, and NSAIDs, by computing the effect of their combination in an additive scale [19]. For that purpose, we built a variable with 5 categories; non-use of any of the two drug classes, current use of statins alone, current use of the other drug class alone, current use of both drug classes combined, and rest of combinations (past and recent users of both); we then used it as the variable of exposure in the logistic model. When we evaluated the association of each individual statin with CRC, we excluded all current users with previous recorded prescriptions of another statin in order to avoid a residual or cumulative effect. Among current users, we additionally examined whether lipophilicity of statins (lipophilic or hydrophilic) and intensity of treatment (low, moderate, or high) modified the main association with CRC.

### 2.7. Sensitivity Analysis

From the analysis, we excluded all users of low-dose aspirin, clopidogrel and NSAIDs ever and in the prior year before index date to avoid a residual effect of other drugs known to reduce the risk of CRC.

### 2.8. Systematic Review and Meta-Analysis

We carried out the review in accordance with the Preferred Reporting Items for Systematic Reviews and Meta-analysis (PRISMA). A systematic literature search was conducted in PubMed and Web of Science for all studies published up to 31 January 2022, without language restriction, and using the following terms: “Statin*”, “Hydroxymethylglutaryl-CoA Reductase Inhibitor”, “3-hydroxy-3-methylglutaryl-coenzyme”, “HMG-CoA”, “Colon cancer risk”, “Rectal cancer risk”, “Colorectal cancer risk”, “Colorectal neoplas* risk”, “Colorectal adenom* risk”. Additionally, we perused the list of references of the studies reviewed to identify potential eligible studies. To consider a study as eligible it must fulfil the following criteria: (1) cohort or case-control designs, (2) aimed to evaluate the association between statin use and the risk of CRC (including colorectal, or colon and rectum separately), (3) provide a measure of association (risk ratio, incidence rate ratio, odds ratio or hazard ratio) adjusted for potential confounders, and their 95% CIs, (4) include at least 50 CRC cases. We excluded: (1) experimental studies and systematic reviews and/or meta-analyses, and (2) all studies including mortality, progression, or prognosis after CRC diagnosis as the sole outcome. Studies conducted only among females or males as well as those that only reported estimates by duration, were also included and pooled. A random effects model was used to produce pooled effect sizes (ES) in the multiplicative scale (including odds ratios, relative risks, incidence rate ratios and hazard ratios) with their 95% CIs. Heterogeneity was measured through the statistic I^2^. All of the results were displayed in a forest plot stratified by study design, and we evaluated the publication bias through a funnel plot.

The level of statistical significance was set at *p* < 0.05. All statistical analyses, including MICE and meta-analysis, were performed with STATA/SE, v.14.2 (Statacorp LLC, College Station, TX, USA).

## 3. Results

From a study cohort of 5,310,198 subjects throughout the study period, we identified and extracted 15,491 incident CRC cases and 60,000 controls successfully matched by age (mean: 68.6, SD ± 11.8, years), sex (males 58.8%) and year of index date (Figure 1). The median follow-up (in years) since start date among cases and controls was 3.05 (interquartile range (IQR): 4.57) and 2.80 (IQR: 4.28), respectively.

### 3.1. Characteristics of Cases and Controls

Characteristics of cases and controls at index date are shown in Table 1. Past smokers, alcohol abusers, and history of diabetes, gout, hypertension, angina pectoris, peripheral artery disease and acute GI disorders (constipation, anorectal pathology, complicated and non-complicated upper GI disorders, dyspepsia and lower GI bleeding) were more prevalent among cases resulting in a crude positive association with CRC, which was still present after full adjustment, excepting for diabetes, hypertension, angina pectoris, constipation, non-complicated upper GI disorders and dyspepsia that lost the statistical significance. On the other hand, history of acute myocardial infarction and inflammatory bowel disease were more prevalent among controls resulting in a crude negative association with CRC, but after a full adjustment also stroke and irritable bowel syndrome showed a significant negative association with CRC risk (Table 1). Current users of statins compared to non-users, among controls, were mostly men, overweight or obese (BMI ≥ 25), aged 70-years or older and with hypertension under treatment with antihypertensive drugs (Appendix A).

### 3.2. Association of Statins with Colorectal Cancer, by Recency, Duration and Subgroup Analyses

Current users of statins were equally prevalent among cases and controls resulting in a crude OR of 0.99 (95% CI: 0.94–1.03), but after full adjustment the AOR was 0.87 (95% CI: 0.83–0.91). Upon discontinuation, the magnitude of the AOR slightly weakened in recent users (AOR = 0.91; 95% CI: 0.83–1.01) and disappeared in past users (AOR = 1.09; 95% CI: 1.00–1.19) (Table 2). Such reduced risk among current users was only significant after 180-days of continuous exposure (AOR = 0.85; 95% CI: 0.76–0.96), then the magnitude of the reduction increased after 1–3 years (AOR = 0.79; 95% CI: 0.73–0.84) though after this, it declined (AOR_>3-years_ = 0.94; 95% CI: 0.87–1.02; Table 2). The exclusion of users of NSAIDs and/or antiplatelet drugs barely changed the estimators (Table 2).

We found a significant reduced risk of CRC associated with statins in both sexes, in patients older and younger than 70-years, and across all categories of BMI. However, the reduced risk was greater in subjects aged 70-years or younger, as compared to the older group (*p*-value for interaction = 0.04; Figure 2).

### 3.3. Association of Statins with CRC by Active Principle, Lipophilicity and Intensity

By active principle, we found a trend of a reduced risk with the current use of all statins, albeit statistically significant results were only reached with simvastatin (AOR = 0.86; 95% CI: 0.80–0.93) and rosuvastatin (AOR = 0.58; 95% CI: 0.41–0.83), being marginally significant for atorvastatin (AOR = 0.92; 95% CI: 0.84–1.00; Table 3).

The association of low-intensity statins with a reduced risk of CRC was of a lesser magnitude (AOR = 0.93; 95% CI: 0.85–1.01) than that observed with moderate (AOR = 0.84; 95% CI: 0.79–0.90) and high (AOR = 0.85; 95% CI: 0.74–0.98) intensities, although itreached the statistical significance when duration of use was longer than 1-year (AOR = 0.89; 95% CI: 0.81–0.99; Table 4). The reduced risk of CRC was observed with lipophilic and hydrophilic statins alike (Table 4).

### 3.4. Potential Interaction of Statins with Other Drugs

The association of statins with a reduced risk of CRC was apparently not modified by their combination with antihypertensives, antiplatelet drugs, other lipid-lowering drugs or NSAIDs, with the exception of fibrates, where the point estimate of the combination was greater than the independent results of each drug class (AOR = 0.61; 95% CI: 0.44–0.84; Table 5).

### 3.5. Systematic Review and Meta-Analysis

The automated search retrieved a total of 341 articles, of which 47 met the inclusion criteria (Appendix A) so finally 48 studies (47 plus the present study) were pooled. Their main characteristics are shown in Appendix A. Individual ES (95% CI), weights (in percentage) and pooled estimates (by study design and overall) are shown in Figure 3. Pooled-ES (95% CI) across strata showed a moderate risk reduction in CRC associated to statins use; among case-control studies ES was 0.90 (95% CI: 0.86–0.96), among cohort studies ES was 0.88 (95% CI: 0.82–0.95), and overall ES was 0.90 (95% CI: 0.86–0.93). All strata showed a high heterogeneity (I^2^ > 80%). The funnel plot (Appendix A) showed asymmetry suggesting a publication bias favouring a protective effect.

## 4. Discussion

Results from the case-control study support the hypothesis that the current use of statins is associated with a moderate risk reduction (around 15%) of developing CRC when the duration of treatment is longer than 180 days. Such reduced risk barely persisted up to 1-year upon discontinuation but not beyond. All of the individual statins examined showed a trend of a reduced risk of variable magnitude without differences regarding lipophilicity, while moderate and high intensity of treatment seems to present a greater effect. The concomitant use of low-dose aspirin, other non-aspirin antiplatelet drugs (as clopidogrel), NSAIDs, antihypertensive drugs, or other lipid-lowering drugs did not modify the main effect of statins, with the exception of fibrates which showed an independent risk reduction that may be additional to the one observed with statins when both of them are combined. Finally, the synthesis of the epidemiological evidence also supports the hypothesis of a moderate risk reduction (10%) of CRC associated to statins use.

Statins are competitive and reversible inhibitors of the HMG-CoA reductase, a key enzyme for the synthesis of cholesterol in the liver. In the pathway from HMG-CoA to cholesterol, numerous intermediate metabolites are generated, contributing to the prenylation of proteins Ras, Rho and Rap, which are involved in several cell signaling functions related to cell growth, proliferation and migration, superoxide generation and oxidative stress, or increased platelet activation [16,24], and some of them are closely concerned with carcinogenesis [14,16,24]. Moreover, CRC cells overexpress HMG-CoA reductase resulting in an increased biosynthesis of mevalonate, cholesterol, Ras and Rho, leading to potential deleterious effects as a consequence [16,24]. In addition, some authors have shown that the use of statins may lead to lower prevalence of gut microbiota dysbiosis and favors the growth of species whose metabolites may exert anti-inflammatory effects as *Bifidobacterium* [25,26,27,28].

Our study presents several features that have been poorly explored in the literature that need to be discussed: first, our study was performed in a Mediterranean population, characterized by different lifestyle factors and lower CV morbidity. A similar study performed in the Spanish region of Catalonia was recently published [29] and authors found no association between statins and CRC, although limited individual validation of the exposure and cancer status, as the authors recognized, may have led to a misclassification that could have distorted the risk estimates towards the null.

Second, the reduced risk of CRC associated with statins was time-dependent and started 6-months after the initiation of treatment. Such an early effect was also observed in previous studies with statins [30,31], but also with other drugs such as low-dose aspirin or NSAIDs [32,33], yet it is a matter of controversy. In addition, we observed that the association of statins with a reduced risk of CRC did not persist upon discontinuation of treatment, suggesting a reversible effect, although additional studies are needed to support this hypothesis. The reduced risk of CRC seems to vanish after 3-years of use, which is an unexpected result. Nevertheless, other authors found a similar trend [34,35] or even an increased long-term risk [36]. In our view, this could be partly explained by a time-dependent lack of adherence to chronic treatments.

Third, we observed that the reduced risk of CRC associated with statins appeared to be higher among subjects aged 70-years or younger, which may be consistent with the idea that younger statin users are more adherent to healthy habits [37,38] such as screening colonoscopy, resulting in a lower risk of CRC [31], or it may truly mean that older subjects have less potential to experience benefits.

Fourth, regarding individual statins, we observed a significant reduced risk of CRC associated to current use of simvastatin, rosuvastatin and marginally to atorvastatin. Although for the rest the statistical significance was not reached, all statins showed a trend to a reduced risk, which can be interpreted as if there were a class effect. The protection observed for rosuvastatin differs substantially from the one observed for the rest of statins. Despite the fact that complex anti-tumor mechanisms have been specifically described for rosuvastatin in in-vivo studies [39], the extent of those in the prevention of CRC is unknown. Furthermore, since pleiotropic effects are a beneficial consequence of the inhibition of HMG-CoA reductase, it would be pharmacologically plausible that the higher the intensity of such inhibition the more enhanced pleiotropic effects [16]. On the other hand, it has been hypothesized that lipophilic statins may have a greater chemoprotective effect as they can cross cell membranes and exert pleiotropic effects in many tissues while hydrophilic statins depend on specific membrane transporters [16]. However, our results do not support this hypothesis.

Fifth, the association of statins with a reduced risk of CRC was observed even when used alone or combined with other drugs for CVD prevention or NSAIDs. Interestingly, our results suggest that the association of statins with fibrates has a greater effect than the one separately observed with each drug, which is compatible with, at least, an additive effect. Fibrates activate the peroxisome proliferator-activated receptors (PPAR) alpha, which beyond its action in lipid homeostasis, have been described as playing a role in the modulation of the inflammatory response and tumorigenesis [40]. In addition, it is described that the biological actions deployed by NSAIDs are not exclusively mediated by COX inhibition but also through the activation of PPARs, among others [40,41].

Some limitations should be mentioned. First, there remains a low probability of misclassification of the outcome (<5%) [34] but likely non-differential with respect to the exposure (the process of validation of CRC diagnoses was blinded to drug prescriptions), which may have led to a minor underestimation of the chemoprotective effect. Second, we only had access to electronic prescriptions, so non-adherent subjects could be misclassified; although presumably non-differential with respect to the outcome, it would have resulted in an underestimation of the main association. Moreover, the possibility of unmeasured confounding factors or selection bias could not be ruled out from observational data.

Finally, the results from the synthesis of available evidence are consistent with those observed in the present study towards a moderate reduced risk of CRC associated to statins use, although pooled results were strongly weighed down by a high heterogeneity across studies.

## 5. Conclusions

Results from the case-control study and meta-analysis support a moderate chemoprotective effect of statins against CRC risk. All in all, the risk reduction seems to be a class effect but heterogeneous across statins. The reduced risk was time-dependent and apparently greater among people aged 70-years or younger, and with moderate and high intensity treatments with statins. The reduced risk was observed when statins were used alone of combined with low-dose aspirin and other CV drugs. The greater risk reduction observed when statins were associated with fibrates is interesting and merits further exploration.

## Figures and Tables

**Figure 1 jcm-11-01528-f001:**
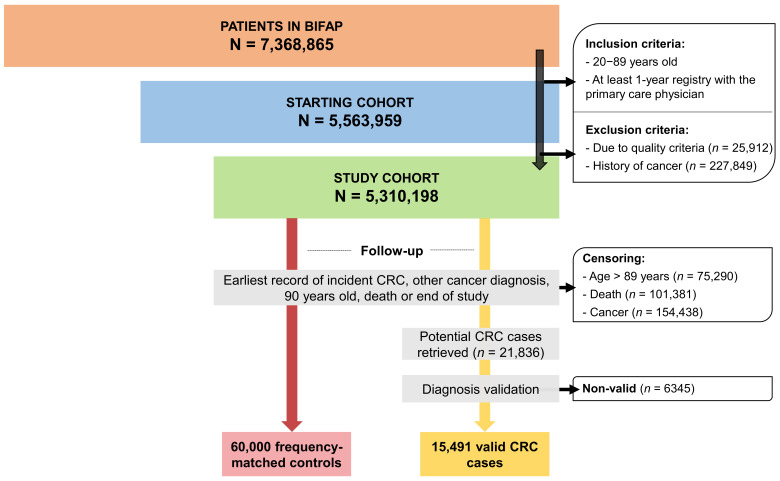
Flowchart of study cohort inception.

**Figure 2 jcm-11-01528-f002:**
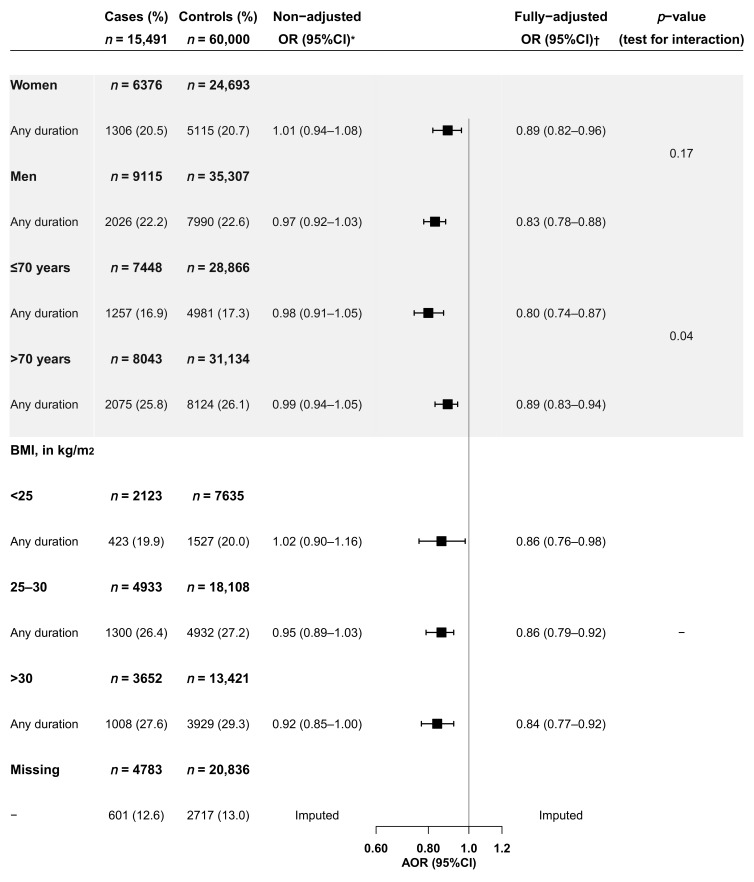
Current use of statins and risk of colorectal cancer, by gender, age and body mass index (BMI). BMI: body mass index; OR: odds ratio; CI: confidence interval * Model adjusted only for matching variables (age, sex, and calendar year). The category of reference was “non-use of statins”. ^†^ Model adjusted for: (1) Matching variables: age, sex and calendar year, (2) Comorbidities and risk factors: number of visits in the last year, BMI, alcohol abuse, smoking, chronic gastritis, reflux, inflammatory bowel disease, irritable bowel syndrome, constipation, anorectal pathology, upper GI disorders, lower GI bleeding, hyperuricemia and gout, and (3) Use of drugs: antihypertensives, low-dose aspirin, non-aspirin antiplatelet drugs, oral anticoagulants, glucose-lowering drugs (oral and insulin), other lipid-lowering drugs, anti-H2 acid suppressors, proton pump inhibitors, antidiarrheal drugs, drugs for constipation, selective serotonin reuptake inhibitors and serotonin receptor antagonists with serotonin reuptake inhibition, analgesic opioids, NSAIDs (non-selective and coxibs), SYSADOAs, calcium and vitamin D supplements, and corticosteroids.

**Figure 3 jcm-11-01528-f003:**
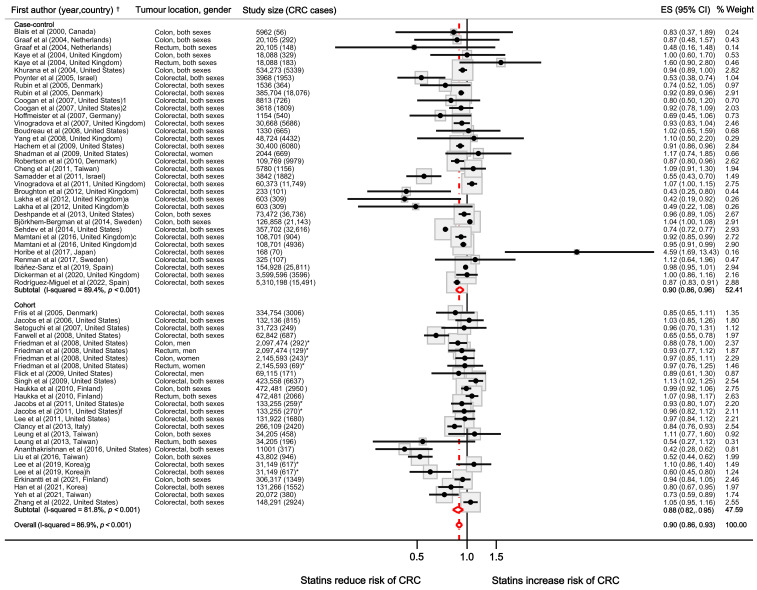
Forest plot. Results from individual studies and pooled estimates from random-effects model. CRC: colorectal cancer; ES: effect size in the multiplicative scale from pooled odds ratios, relative risks, incidence rate ratios and hazard ratios; CI: confidence interval. ^a^ ES for ≥2 dispensed statins prescriptions at least 2 months pre-recruitment. Only reported the total number of CRC cases (not disaggregated). ^b^ ES for ≥2 dispensed statins prescriptions at least 7 months pre-recruitment. Only reported the total number of CRC cases (not disaggregated). ^c^ ES for short-term statin use (<1 year). ^d^ ES for long-term statin use (>1 year). ^e^ ES for <5 years of statin use. ^f^ ES for >5 years of statin use. ^g^ ES for low users (based on Medication Possession Ratio). Only reported the total number of CRC cases (not disaggregated). ^h^ ES for high users (based on Medication Possession Ratio). Only reported the total number of CRC cases (not disaggregated). * Only reported the number of exposed CRC cases. ^†^ See Appendix A for references in the figure.

**Table 1 jcm-11-01528-t001:** Baseline characteristics of cases and controls.

	Cases *n* = 15,491	Controls *n* = 60,000	Non-Adjusted OR * (95% CI)	Fully-Adjusted-OR ^†^ (95% CI)
Age, mean (±SD), years	68.6 (11.8)	68.6 (11.8)	Matched	Matched
Women, *n* (%)	6376 (41.2)	24,693 (41.2)	Matched	Matched
Follow-up in years, median (IQR)	3.05 (4.57)	2.80 (4.28)	1.03 (1.02–1.04)	1.04 (1.03–1.05)
Visits to primary care in the prior year, *n* (%):				
<6	3248 (21.0)	19,508 (32.5)	Reference	Reference
6–10	3639 (23.5)	13,357 (22.3)	1.71 (1.62–1.80)	1.83 (1.73–1.93)
11–20	5298 (34.2)	17,118 (28.5)	2.01 (1.91–2.11)	2.23 (2.11–2.36)
>20	3306 (21.3)	10,017 (16.7)	2.23 (2.11–2.37)	2.52 (2.35–2.70)
BMI, kg/m^2^, *n* (%):				
<24.99	2123 (13.7)	7635 (12.7)	Reference	Reference
25–30	4933 (31.8)	18,108 (30.2)	1.00 (0.95–1.05)	0.98 (0.93–1.03)
>30	3652 (23.6)	13,421 (22.4)	0.99 (0.93–1.05)	0.97 (0.91–1.03)
Missing	4783 (30.9)	20,836 (34.7)	Imputed	Imputed
Smoking, *n* (%):				
Non-smoker	4904 (31.7)	18,016 (30.0)	Reference	Reference
Current smoker	2384 (15.4)	8960 (14.9)	0.99 (0.92–1.06)	1.04 (0.97–1.12)
Past smoker	1217 (7.86)	3821 (6.37)	1.19 (1.10–1.28)	1.19 (1.11–1.29)
Missing	6986 (45.1)	29,203 (48.7)	Imputed	Imputed
History of, *n* (%):				
Alcohol abuse ^‡^	552 (3.56)	1578 (2.63)	1.38 (1.25–1.52)	1.24 (1.12–1.38)
Diabetes	3170 (20.5)	10,778 (18.0)	1.18 (1.13–1.23)	1.06 (0.98–1.16)
Non-gout hyperuricemia	984 (6.35)	3829 (6.38)	1.01 (0.93–1.08)	0.92 (0.86–1.00)
Gout	718 (4.63)	2363 (3.94)	1.19 (1.09–1.30)	1.12 (1.02–1.22)
Hypertension	7527 (48.6)	28,051 (46.8)	1.09 (1.05–1.13)	0.98 (0.94–1.04)
Perypheral artery disease	466 (3.01)	1420 (2.37)	1.28 (1.15–1.43)	1.17 (1.04–1.31)
Acute myocardial infarction	502 (3.24)	2213 (3.69)	0.88 (0.79–0.97)	0.86 (0.77–0.96)
Angina pectoris	467 (3.01)	1600 (2.67)	1.13 (1.02–1.26)	1.10 (0.98–1.23)
Stroke ^§^	559 (3.61)	2249 (3.75)	0.96 (0.87–1.06)	0.87 (0.79–0.97)
Transient ischemic attack	281 (1.81)	1109 (1.85)	0.98 (0.86–1.12)	0.94 (0.82–1.08)
Chronic gastritis	154 (0.99)	604 (1.01)	0.99 (0.83–1.18)	0.84 (0.70–1.01)
Gastroesophageal reflux	1866 (12.1)	6896 (11.5)	1.05 (1.00–1.11)	0.93 (0.87–0.98)
Inflammatory bowel disease	51 (0.33)	238 (0.40)	0.83 (0.61–1.12)	0.41 (0.30–0.57)
Irritable bowel syndrome	238 (1.54)	941 (1.57)	0.98 (0.85–1.13)	0.84 (0.72–0.97)
Constipation	1601 (10.3)	5424 (9.04)	1.16 (1.10–1.24)	0.93 (0.81–1.06)
Anorectal pathology ^¶^	1995 (12.9)	5741 (9.57)	1.40 (1.32–1.48)	1.24 (1.17–1.31)
Complicated upper GI disorders **	459 (2.96)	1062 (1.77)	1.73 (1.55–1.94)	1.41 (1.26–1.59)
Non-complicated upper GI disorders ^††^	1103 (7.12)	3889 (6.48)	1.14 (1.06–1.22)	1.02 (0.95–1.10)
Dyspepsia	1797 (11.6)	6549 (10.9)	1.10 (1.04–1.16)	0.97 (0.92–1.03)
Lower GI bleeding	898 (5.80)	1310 (2.18)	2.76 (2.53–3.01)	2.44 (2.23–2.67)

OR: odds ratio; CI: confidence interval; SD: standard deviation; IQR: interquartile range; BMI: body mass index; GI: gastrointestinal. * Model adjusted only for matching variables (age, sex and calendar year). ^†^ Model adjusted for: (1) Matching variables: age, sex and calendar year, (2) Comorbidities and risk factors: number of visits in the last year, BMI, alcohol abuse, smoking, chronic gastritis, reflux, inflammatory bowel disease, irritable bowel syndrome, constipation, anorectal pathology, upper GI disorders, lower GI bleeding, hyperuricemia and gout, and (3) Use of drugs: antihypertensives, low-dose aspirin, non-aspirin antiplatelet drugs, oral anticoagulants, glucose-lowering drugs (oral and insulin), other lipid-lowering drugs, anti-H2 acid suppressors, proton pump inhibitors, antidiarrheal drugs, drugs for constipation, selective serotonin reuptake inhibitors and serotonin receptor antagonists with serotonin reuptake inhibition, analgesic opioids, NSAIDs (non-selective and coxibs), SYSADOAs, calcium and vitamin D supplements, and corticosteroids. The category of reference was “no presence of the disease”. ^‡^ When the general practitioner recorded an excessive consumption of alcohol. ^§^ Includes haemorrhagic and ischemic stroke. ^¶^ Includes hemorrhoids, anal fissure, and anorectal abscess. ** Includes complicated ulcer, gastritis or duodenitis with bleeding and upper GI bleeding. ^††^ Includes non-bleeding or non-complicated ulcer, gastritis or duodenitis.

**Table 2 jcm-11-01528-t002:** Use of statins and risk of colorectal cancer.

	Cases *n* = 15,491	Controls *n* = 60,000	Non-Adjusted OR * (95% CI)	Fully-Adjusted OR ^†^ (95% CI)
Non-users	10,826 (69.9)	42,008 (70.0)	Reference	Reference
Recency of use, in days, *n* (%):				
Current (0–90)	3332 (21.5)	13,105 (21.8)	0.99 (0.94–1.03)	0.87 (0.83–0.91)
Recent (91–365)	540 (3.49)	2068 (3.45)	1.01 (0.92–1.12)	0.91 (0.83–1.01)
Past (>365)	793 (5.12)	2819 (4.70)	1.09 (1.01–1.18)	1.09 (1.00–1.19)
Continuous duration, among current users:				
≤1 year:	1055 (6.81)	4006 (6.68)	1.02 (0.95–1.10)	0.91 (0.84–0.98)
<91 days	409 (2.64)	1575 (2.62)	1.01 (0.90–1.13)	0.94 (0.84–1.05)
91–180 days	249 (1.61)	924 (1.54)	1.05 (0.91–1.20)	0.95 (0.82–1.10)
181 days–1 year	397 (2.56)	1507 (2.51)	1.02 (0.91–1.14)	0.85 (0.76–0.96)
>1 year:	2277 (14.7)	9099 (15.2)	0.97 (0.92–1.02)	0.85 (0.81–0.90)
366 days–3 years	1195 (7.71)	5081 (8.47)	0.91 (0.85–0.98)	0.79 (0.73–0.84)
>3 years	1082 (6.98)	4018 (6.70)	1.04 (0.97–1.12)	0.94 (0.87–1.02)
				*p* for trend < 0.001
Excluding prior use of NSAIDs ^‡^ and/or antiplatelet drugs, ^§^ among current users:				
Ever:	*n* = 4646	*n* = 18,086		
Any duration	558 (12.0)	2229 (12.3)	0.98 (0.89–1.08)	0.79 (0.71–0.88)
Continuous duration:				
≤1 year	187 (4.02)	805 (4.45)	0.91 (0.77–1.07)	0.77 (0.65–0.92)
>1 year	371 (7.99)	1424 (7.87)	1.02 (0.90–1.15)	0.80 (0.71–0.91)
In the prior year:	*n* = 8826	*n* = 33,277		
Any duration	1211 (13.7)	4520 (13.6)	1.01 (0.94–1.08)	0.82 (0.76–0.89)
Continuous duration:				
≤1 year	411 (4.66)	1518 (4.56)	1.02 (0.91–1.15)	0.85 (0.76–0.96)
>1 year	800 (9.06)	3002 (9.02)	1.00 (0.92–1.09)	0.81 (0.74–0.88)

OR: odds ratio; CI: confidence interval; NSAIDs: nonsteroidal anti-inflammatory drugs. * Model adjusted only for matching variables (age, sex, and calendar year). ^†^ Model adjusted for: (1) Matching variables: age, sex and calendar year, (2) Comorbidities and risk factors: number of visits in the last year, BMI, alcohol abuse, smoking, chronic gastritis, reflux, inflammatory bowel disease, irritable bowel syndrome, constipation, anorectal pathology, upper GI disorders, lower GI bleeding, hyperuricemia and gout, and (3) Use of drugs: antihypertensives, low-dose aspirin, non-aspirin antiplatelet drugs, oral anticoagulants, glucose-lowering drugs (oral and insulin), other lipid-lowering drugs, anti-H2 acid suppressors, proton pump inhibitors, antidiarrheal drugs, drugs for constipation, selective serotonin reuptake inhibitors and serotonin receptor antagonists with serotonin reuptake inhibition, analgesic opioids, NSAIDs (non-selective and coxibs), SYSADOAs, calcium and vitamin D supplements, and corticosteroids. ^‡^ Including non-aspirin NSAIDs (COX-2 selective and non-selective). ^§^ Including low-dose aspirin, cilostazol, clopidogrel, dypiridamole, ditazole, prasugrel, ticagrelor, ticlopidine and triflusal.

**Table 3 jcm-11-01528-t003:** Current use of individual statins and risk of colorectal cancer.

	Cases *n* = 15,491	Controls *n* = 60,000	Non-Adjusted OR * (95% CI)	Fully-Adjusted OR ^†^ (95% CI)
**Non-users**	10,826 (69.9)	42,008 (70.0)	Reference	Reference
**Any duration:**				
Simvastatin	1097 (8.85)	4310 (8.97)	0.99 (0.92–1.06)	0.86 (0.80–0.93)
Atorvastatin	951 (7.85)	3642 (7.76)	1.01 (0.94–1.09)	0.92 (0.84–1.00)
Pravastatin	253 (2.26)	966 (2.23)	1.02 (088–1.17)	0.92 (0.80–1.06)
Lovastatin	177 (1.60)	684 (1.59)	1.00 (0.85–1.19)	0.96 (0.81–1.13)
Fluvastatin	117 (1.06)	448 (1.05)	1.01 (0.82–1.24)	0.91 (0.74–1.13)
Rosuvastatin	37 (0.34)	221 (0.52)	0.65 (0.46–0.92)	0.58 (0.41–0.83)
Pitavastatin	10 (0.09)	35 (0.08)	1.11 (0.55–2.24)	0.93 (0.46–1.91)

OR: odds ratio; CI: confidence interval. * Model adjusted only for matching variables (age, sex, and calendar year). The category of reference was “non-use of statins”. ^†^ Model adjusted for: (1) Matching variables: age, sex and calendar year, (2) Comorbidities and risk factors: number of visits in the last year, BMI, alcohol abuse, smoking, chronic gastritis, reflux, inflammatory bowel disease, irritable bowel syndrome, constipation, anorectal pathology, upper GI disorders, lower GI bleeding, hyperuricemia and gout, and (3) Use of drugs: antihypertensives, low-dose aspirin, non-aspirin antiplatelet drugs, oral anticoagulants, glucose-lowering drugs (oral and insulin), other lipid-lowering drugs, anti-H2 acid suppressors, proton pump inhibitors, antidiarrheal drugs, drugs for constipation, selective serotonin reuptake inhibitors and serotonin receptor antagonists with serotonin reuptake inhibition, analgesic opioids, NSAIDs (non-selective and coxibs), SYSADOAs, calcium and vitamin D supplements, and corticosteroids.

**Table 4 jcm-11-01528-t004:** Current use of statins and risk of colorectal cancer, by intensity and lipophilicity.

	Cases *n* = 15,491	Controls *n* = 60,000	Non-Adjusted OR * (95% CI)	Fully-Adjusted OR ^†^ (95% CI)
**Non-users**	10,826 (69.9)	42,008 (70.0)	Reference	Reference
**Intensity: ^‡^**				
Any duration:				
Low intensity	747 (4.82)	2748 (4.58)	1.05 (0.97–1.15)	0.93 (0.85–1.01)
Moderate intensity	1714 (11.1)	6855 (11.4)	0.97 (0.92–1.03)	0.84 (0.79–0.90)
High intensity	285 (1.84)	1103 (1.84)	1.00 (0.88–1.14)	0.85 (0.74–0.98)
Missing dose	586 (3.78)	2399 (4.00)	0.95 (0.86–1.04)	0.88 (0.80–0.97)
Continuous duration:				
Low intensity:				
≤1 year	235 (1.52)	793 (1.32)	1.15 (0.99–1.33)	1.02 (0.88–1.18)
>1 year	512 (3.31)	1955 (3.26)	1.02 (0.92–1.12)	0.89 (0.81–0.99)
Moderate intensity:				
≤1 year	536 (3.46)	2083 (3.47)	1.00 (0.91–1.10)	0.87 (0.78–0.96)
>1 year	1178 (7.60)	4772 (7.95)	0.96 (0.89–1.02)	0.83 (0.77–0.89)
High intensity:				
≤1 year	87 (0.56)	332 (0.55)	1.02 (0.80–1.29)	0.89 (0.70–1.14)
>1 year	198 (1.28)	771 (1.29)	1.00 (0.85–1.17)	0.83 (0.70–0.98)
**Lipophilicity:**				
Any duration:				
Lipophilic ^§^	2947 (19.0)	11,503 (19.2)	0.99 (0.95–1.04)	0.88 (0.83–0.92)
Hydrophilic ^¶^	385 (2.49)	1604 (2.67)	0.93 (0.83–1.04)	0.81 (0.72–0.91)
Continuous duration:				
Lipophilic:				
≤1 year	933 (6.02)	3536 (5.89)	1.02 (0.95–1.10)	0.91 (0.84–0.98)
>1 year	2014 (13.0)	7965 (13.3)	0.98 (0.93–1.03)	0.86 (0.81–0.92)
Hydrophilic:				
≤1 year	122 (0.79)	470 (0.78)	1.01 (0.82–1.23)	0.88 (0.72–1.08)
>1 year	263 (1.70)	1134 (1.89)	0.90 (0.78–1.03)	0.78 (0.68–0.90)

OR: odds ratio; CI: confidence interval. * Model adjusted only for matching variables (age, sex and calendar year). The category of reference was “non-use of statins”. ^†^ Model adjusted for: (1) Matching variables: age, sex and calendar year, (2) Comorbidities and risk factors: number of visits in the last year, BMI, alcohol abuse, smoking, chronic gastritis, reflux, inflammatory bowel disease, irritable bowel syndrome, constipation, anorectal pathology, upper GI disorders, lower GI bleeding, hyperuricemia and gout, and (3) Use of drugs: antihypertensives, low-dose aspirin, non-aspirin antiplatelet drugs, oral anticoagulants, glucose-lowering drugs (oral and insulin), other lipid-lowering drugs, anti-H2 acid suppressors, proton pump inhibitors, antidiarrheal drugs, drugs for constipation, selective serotonin reuptake inhibitors and serotonin receptor antagonists with serotonin reuptake inhibition, analgesic opioids, NSAIDs (non-selective and coxibs), SYSADOAs, calcium and vitamin D supplements, and corticosteroids. ^‡^ Intensity according to American Heart Association [20]: low-intensity (simvastatin < 20 mg, fluvastatin 20–40 mg, lovastatin 20 mg, pitavastatin 1 mg, and pravastatin ≤ 20 mg); moderate-intensity (simvastatin 20–40 mg, fluvastatin 40–80 mg, lovastatin 40 mg, pitavastatin 2–4 mg, pravastatin 40–80 mg, rosuvastatin 5–10 mg, and atorvastatin 10–20 mg); high-intensity (atorvastatin 40–80 mg and rosuvastatin 20–40 mg). ^§^ Includes simvastatin, atorvastatin, pitavastatin, fluvastatin, lovastatin, and cerivastatin. ^¶^ Includes rosuvastatin and pravastatin.

**Table 5 jcm-11-01528-t005:** Interactions between current use of statins and current use of cardiovascular drugs or nonsteroidal anti-inflammatory drugs (NSAIDs), and risk of colorectal cancer.

	Cases *n* = 15,491	Controls *n* = 60,000	Non-Adjusted OR * (95% CI)	Fully-Adjusted OR ^†^ (95% CI)
**Antihypertensive drugs:**				
Non-users	10,453 (67.5)	40,572 (67.6)	Ref.	Ref.
Statins only	3121 (20.2)	12,273 (20.5)	0.99 (0.94–1.03)	0.87 (0.82–0.91)
Alpha blockers only	199 (1.28)	895 (1.49)	0.86 (0.74–1.01)	0.76 (0.65–0.89)
Statins + Alpha blockers	141 (0.91)	484 (0.81)	1.13 (0.94–1.37)	0.97 (0.80–1.18)
Non-users	9834 (63.5)	38,316 (63.9)	Ref.	Ref.
Statins only	2399 (15.5)	9488 (15.8)	0.99 (0.94–1.04)	0.86 (0.81–0.91)
Beta blockers only	644 (4.16)	2263 (3.77)	1.11 (1.01–1.21)	0.94 (0.85–1.03)
Statins + Beta blockers	744 (4.80)	2901 (4.83)	0.99 (0.92–1.09)	0.84 (0.76–0.92)
Non-users	9765 (63.0)	38,146 (63.6)	Ref.	Ref.
Statins only	2589 (16.7)	10,252 (17.1)	0.99 (0.94–1.04)	0.86 (0.82–0.91)
ARBs only	647 (4.18)	2306 (3.84)	1.10 (1.00–1.20)	0.92 (0.84–1.01)
Statins + ARBs	484 (3.12)	1851 (3.08)	1.02 (0.92–1.13)	0.84 (0.76–0.94)
Non-users	8906 (57.5)	35,050 (58.4)	Ref.	Ref.
Statins only	2076 (13.4)	8449 (14.1)	0.97 (0.92–1.02)	0.84 (0.80–0.89)
ACEIs only	1033 (6.67)	3829 (6.38)	1.06 (0.99–1.15)	0.88 (0.81–0.95)
Statins + ACEIs	788 (5.09)	2917 (4.95)	1.05 (0.96–1.14)	0.83 (0.76–0.91)
Non-users	9566 (61.8)	37,225 (62.0)	Ref.	Ref.
Statins only	2424 (15.7)	9623 (16.0)	0.98 (0.93–1.03)	0.85 (0.81–0.90)
CCBs only	758 (4.89)	2916 (4.86)	1.01 (0.93–1.10)	0.89 (0.82–0.97)
Statins + CCBs	625 (4.03)	2440 (4.07)	1.00 (0.91–1.09)	0.84 (0.76–0.93)
Non-users	9570 (61.8)	37,309 (62.2)	Ref.	Ref.
Statins only	2529 (16.3)	10,081 (16.8)	0.98 (0.93–1.03)	0.85 (0.80–0.90)
Diuretics only ^‡^	839 (5.42)	3201 (5.33)	1.02 (0.94–1.11)	0.88 (0.81–0.96)
Statins + diuretics	545 (3.52)	2269 (3.78)	0.94 (0.85–1.03)	0.78 (0.71–0.87)
**Antiplatelet drugs:**				
Non-users	9626 (62.1)	37,319 (62.2)	Ref.	Ref.
Statins only	1970 (12.7)	7942 (13.2)	0.96 (0.91–1.02)	0.83 (0.78–0.88)
Low-dose aspirin only	712 (4.60)	2928 (4.88)	0.94 (0.86–1.03)	0.81 (0.74–0.88)
Statins + Low-dose aspirin	1064 (6.87)	4212 (7.02)	0.98 (0.91–1.05)	0.80 (0.74–0.87)
Non-users	10,671 (68.9)	41,465 (69.1)	Ref.	Ref.
Statins only	2971 (19.2)	11,701 (19.5)	0.99 (0.94–1.03)	0.87 (0.83–0.92)
Clopidogrel only	107 (0.69)	386 (0.64)	1.08 (0.87–1.34)	0.82 (0.66–1.03)
Statins + Clopidogrel	229 (1.48)	927 (1.54)	0.96 (0.83–1.11)	0.75 (0.65–0.88)
**Other lipid-modifying agents:**				
Non-users	10,792 (69.7)	41,848 (69.8)	Ref.	Ref.
Statins only	3206 (20.7)	12,577 (21.0)	0.99 (0.94–1.03)	0.87 (0.82–0.91)
Other lipid-lowering drugs only ^§^	21 (0.14)	90 (0.15)	0.90 (0.56–1.46)	0.70 (0.43–1.15)
Statins + Other lipid-lowering drugs	75 (0.48)	314 (0.52)	0.93 (0.72–1.19)	0.80 (0.62–1.04)
Non-users	10,612 (68.5)	41,143 (68.6)	Ref.	Ref.
Statins only	3154 (20.4)	12,383 (20.6)	0.99 (0.94–1.03)	0.87 (0.83–0.91)
Fibrates only	113 (0.73)	485 (0.81)	0.90 (0.74–1.11)	0.79 (0.64–0.98)
Statins + Fibrates	47 (0.30)	246 (0.41)	0.74 (0.54–1.01)	0.61 (0.44–0.84)
**NSAIDs: ^¶^**				
Non-users	4356 (28.1)	16,996 (28.3)	Ref.	Ref.
Statins only	1074 (6.93)	4227 (7.04)	0.99 (0.92–1.07)	0.82 (0.76–0.89)
NSAIDs only	1433 (9.25)	6567 (11.0)	0.85 (0.80–0.91)	0.66 (0.61–0.71)
Statins + NSAIDs	581 (3.75)	2632 (4.39)	0.86 (0.78–0.95)	0.65 (0.58–0.72)

NSAIDs: nonsteroidal anti-inflammatory drugs; OR: odds ratio; CI: confidence interval; ARB: angiotensin-II receptor blockers; ACEI: angiotensin-converting enzyme inhibitors; CCB: calcium channel blockers. * Model adjusted only for matching variables (age, sex, and calendar year). The category of reference was “non-use of both drug classes”. ^†^ Model adjusted for: (1) Matching variables: age, sex and calendar year, (2) Comorbidities and risk factors: number of visits in the last year, BMI, alcohol abuse, smoking, chronic gastritis, reflux, inflammatory bowel disease, irritable bowel syndrome, constipation, anorectal pathology, upper GI disorders, lower GI bleeding, hyperuricemia and gout, and (3) Use of drugs: antihypertensives, low-dose aspirin, non-aspirin antiplatelet drugs, oral anticoagulants, glucose-lowering drugs (oral and insulin), other lipid-lowering drugs, anti-H2 acid suppressors, proton pump inhibitors, antidiarrheal drugs, drugs for constipation, selective serotonin reuptake inhibitors and serotonin receptor antagonists with serotonin reuptake inhibition, analgesic opioids, NSAIDs (non-selective and coxibs), SYSADOAs, calcium and vitamin D supplements, and corticosteroids. ^‡^ Includes hydrochlorothiazide in combination with ARBs or ACEIs. ^§^ Includes bile-acid sequestrants and ezetimibe. ^¶^ Includes COX-2 selective and non-selective NSAIDs.

## Data Availability

The code to reproduce all the analyses is publicly available in: https://github.com/antoniohupa/statin_use_and_CRC_risk. At this moment, the use of data from BIFAP is restricted to researchers from non-profit institutions, so the anonymized dataset will be only available on informed request from the corresponding author.

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
