# Peer review of "Statins and Colorectal Cancer Risk: A Population-Based Case-Control Study and Synthesis of the Epidemiological Evidence"

_jcm, 2022, doi:10.3390/jcm11061528_

Round 1
Reviewer 1 Report
This manuscript (JCM-1574521) is an interesting study elucidating the effect of statins against colorectal cancer through a case-control study and meta-analysis and suggested statins have chemoprotective efficacy. This study highlights the clinical insights of statins in subjects with CRC. However, there are several concerns listed below.
- It would be a great addition to the study if the authors provided the multivariate analysis of the baseline characteristics and how these parameters correlate with statins in patients with CRC.
- Please provide the information about the institutional review boards.
- It is unclear how the authors calculated the sample size to provide sufficient study power.
- What criteria were used to diagnose irritable bowel syndrome/inflammatory bowel disease patients? Are these clinical phenotypes associated with CRC outcomes? The rationale is not clear why the authors include these parameters.
- It is not clear at what time point after statin therapy there was a significant improvement in CRC clinical outcome (symptoms improvement). A correlation analysis is warranted for a better presentation of the data.
- What is the survival rate of the patients with CRC after statin therapy? Did the authors follow up on these patients for survival rate?
- The title of the article should be shortened with a precise conclusion.
- The discussion section needs extensive revision, particularly citing some studies that do not support statin use's benefit in CRC chemoprevention (PMID: 34730560). Further discussion about the alterations in gut microbiota dynamics in statin users with CRC.
- The authors should address the limitations of the study.
- Further attention to language and organization would enhance the manuscript.
Author Response
"Please see the attachment."

Reviewer 2 Report
The current cohort study examined the effect of statins on the incidence of CRC. The authors found that based on their findings and the results of up-dated meta-analysis, the statins had a moderate chemo-preventive effect of stains on CRC risk. The authors should be congratulated for this well-performed study. I have only one minor question:
In the current study, the authors found that the effect is dependent on duration, intensity, and age. However, in updated meta-analysis, no subgroup analysis was performed. What is the association of statins with CRC by recency, duration, subgroup, active principle, lipophilicity and intensity in meta-analysis? This will provide more information about the topic.
Author Response
"Please see the attachment."

Reviewer 3 Report
In this manuscript, the authors revealed a moderate chemoprotective effect of statins against CRC, which was time-dependent and apparently greater among people younger than 70 years old, and with moderate and high intensity statins.
The chemoprotective effect of statins against CRC has remained a mystery. Based on the primary healthcare database BIFAP, the authors built a cohort and extracted 15,491 CRC cases and 60,000 matched controls. To estimate the association between the exposure of statins and CRC, they computed the adjusted‐odds ratios (AOR) by a logistic regression analysis, and carried out a systematic review and updated meta-analysis.
This paper indicated that the effect of statins on the risk of CRC depends on duration, intensity and age. The data is clear and clinically significant.
Minor concerns:
- Table 5
The effect of combination statins with NASIDs needs to be further explained, where the point estimate of the combination was greater than the independent results of each drug class.
- Figure 3
High heterogeneity(I2 d>80%) of meta-analysis is not convincing, it would be better to perform subgroup analyses according to gender, age, race, stage of tumor and so on.
3.Discussion
The paragraphs of discussion are out of order. The fifth is missing.
Author Response
"Please see the attachment."
